# Transcriptome profiling of laser-captured crown root primordia reveals new pathways activated during early stages of crown root formation in rice

Jérémy Lavarenne[1,2], Mathieu Gonin[1], Antony Champion[1], Marie Javelle[2], Hélène Adam[1], Jacques Rouster[2], Geneviève Conejéro[3,4], Marc Lartaud[3,4], Jean-Luc Verdeil[3,4], Laurent Laplaze[1], Christophe Sallaud[2], Mikael Lucas[1], Pascal Gantet[1] *

1 Université de Montpellier, IRD, UMR DIADE, Montpellier, France, 2 Limagrain Field Seeds, Traits and Technologies, Groupe Limagrain—Centre de Recherche, Route d'Ennezat, Chappes, France, 3 CIRAD, UMR1334 AGAP, PHIV-MRI, Montpellier, France, 4 Université de Montpellier, CIRAD, INRA, Montpellier SupAgro, Montpellier, France

* pascal.gantet@umontpellier.fr

**Data Availability Statement:** The datasets supporting the conclusions of this article have been deposited in NCBI's Gene Expression Omnibus and

## Abstract

Crown roots constitute the main part of the rice root system. Several key genes involved in crown root initiation and development have been identified by functional genomics approaches. Nevertheless, these approaches are impaired by functional redundancy and mutant lethality. To overcome these limitations, organ targeted transcriptome analysis can help to identify genes involved in crown root formation and early development. In this study, we generated an atlas of genes expressed in developing crown root primordia in comparison with adjacent stem cortical tissue at three different developmental stages before emergence, using laser capture microdissection. We identified 3975 genes differentially expressed in crown root primordia. About 30% of them were expressed at the three developmental stages, whereas 10.5%, 19.5% and 12.8% were specifically expressed at the early, intermediate and late stages, respectively. Sorting them by functional ontology highlighted an active transcriptional switch during the process of crown root primordia formation. Cross-analysis with other rice root development-related datasets revealed genes encoding transcription factors, chromatin remodeling factors, peptide growth factors, and cell wall remodeling enzymes that are likely to play a key role during crown root primordia formation. This atlas constitutes an open primary data resource for further studies on the regulation of crown root initiation and development.

## Introduction

Crown roots (CR) are a type of adventitious roots making up most of the root system in rice [1,2]. These roots develop post-embryonically from the stem and play an important role in the adaptation of plants to the soil hydro-mineral status [3,4]. Genetics and functional genomics approaches revealed several genes involved in CR development in rice [5,6]. However, these approaches are impaired by functional redundancies, and by lethality of some mutants.

are accessible through GEO Series accession number GSE133593.

**Funding:** In addition to institutional funding, this work is supported by the French Agence Nationale de la Recherche (Fondation Agropolis, Program Investissement d'Avenir Labex Agro no. ANR-10-LABX-0001-01, 2016–2018, by the Partenariat de Recherche Collaborative Entreprise MASTEROOT no. ANR-17-CE20-0028-01, 2018–2021) and by the CGIAR Research Program (CRP) on rice-agrifood systems (RICE, 2017–2022). J.L. is supported by a CIFRE fellowship (No: 2015/0195) from the Association Nationale de la Recherche Technologique, France, and a financial support from of the seed company Limagrain. Limagrain provided support in the form of salaries for authors J.L., M.J., J.R. and C.S. The specific contribution of these authors are mentioned in the 'author contributions' section. The imaging facility MRI, which is part of the UMS BioCampus Montpellier and a member of the national infrastructure France-BioImaging, supported by the French National Research Agency (ANR-10-INBS-04), provided access to biphoton microscopy and Laser Capture Microdissection facility. The funders had no role in study design, data collection and analysis, decision to publish, or preparation of the manuscript.

**Competing interests:** J.L. is supported by a CIFRE fellowship (No: 2015/0195) from the Association Nationale de la Recherche Technologique, France, and a financial support from Biogemma, a subsidiary of the the seed company Limagrain. J.L., M.J., J.R. and C.S. are employees of the seed company Limagrain. This does not alter our adherence to PLOS ONE policies on sharing data and materials. There are no patents, products in development or marketed products associated with this research to declare.

**Abbreviations:** CK, cytokinin; CR, crown root; DEG, differentially expressed gene; FC, fold change; GA, gibberellin; IAA, auxin; LCM, laser capture microdissection; TF, transcription factor.

Transcriptomic approaches, based on the quantification of gene expression at whole-genome level, overcome these limitations by identifying all the genes specifically expressed during a biological process. This type of approach led to the identification of new genes involved in root development and root physiology in *Arabidopsis* [7–9]. Until now, most of the rice root transcriptome datasets are based on the comparison at the whole root systems level between stressed/unstressed material, developmental stages, genetic backgrounds (knock out mutant or overexpressing lines *versus* wild type) or different root types or root tissues. For instance, transcriptomic analysis of root response to soil potassium deficiency revealed the importance of an ion transporter and several protein kinases encoding genes in the adaptation of the plant to this constraint [10]. Comparison between CR, large and fine lateral roots revealed a significant enrichment in transcripts associated with secondary cell wall metabolism, as well as increased levels of transcripts encoding auxin-signaling related genes in CR [11]. Recently, exogenous hormonal treatments and transcriptome analysis of rice stem base allowed the identification of a new gene involved in CR development [12].

We previously used the conditional ectopic induction of the CROWN ROOTLESS 1 transcription factor in its respective mutant background to control the formation of CR in stem bases and generate a transcriptomic dataset capturing the kinetics of gene expression during very early stages of CR formation [13,14]. These approaches are powerful but lack spatial resolution. High spatial resolution transcriptomic analyses can be obtained using laser capture microdissection (LCM). For instance, Jiao et coll. produced a cell type transcriptome atlas that includes 40 cell types from rice shoot, root and germinating seed at several developmental stages [15]. One of the most up-to-date root-based spatialized dataset focused on dividing the CR into eight developmental stages along the longitudinal axis, and three radial tissue types namely: (i) epidermis, exodermis and sclerenchyma; (ii) cortex; (iii) endodermis, pericycle and stele [16]. However, while this dataset covers different tissues of emerged CR, it does not cover the early stages of CR development.

In this paper, with the aim to complete our knowledge about genes expressed during CR formation, we used laser capture microdissection (LCM) to generate an atlas of genes differentially expressed during three early stages of CR primordia development before CR emergence. Using this data resource, transcriptome cross analysis identified key genes already known to be involved in CR initiation and development, and pinpointed new candidate genes involved in gene expression regulation, signaling pathways and cell differentiation and likely controlling CR development.

# Materials and methods

## Plant material and growth conditions

*Oryza sativa* L. *ssp. Japonica* cv. Taichung 65 seeds were disinfected (3 min in ethanol 70%, 90 min in sodium hypochlorite 3.8%:diluted in Milli-Q water plus 500 0L of Tween® 80 (Merck, Germany), quadruple rinsed in sterile Milli-Q water) and germinated under axenic conditions in round Petri dishes (Sarstedt, Germany) containing a filter pad (Whatman paper, GE Healthcare, UK) and 15 mL half strength Murashige and Skoog (MS/2) medium (Duchefa Biochemie, The Netherlands). Culture chambers were set at 27°C, 70% relative humidity, and 14 hours daylight. After 5 days, the plantlets were transferred into 250 mL wide-collar Erlenmeyer flasks containing 30 mL MS/2 medium, and let to grow during 3 days before sampling.

## Stem base sampling, two-photon imaging, tissue preparation, laser capture microdissection and RNA extraction

For two-photon imaging, stem bases were sampled and immersed in MS/2 medium at 4°C overnight. They were included into 5% w:v agarose in sterile water (Duchefa Biochemie, The

Netherlands). Agarose blocks were mounted into the Automate 880, a custom machine combining a LSM NLO 880 multiphoton microscope (Zeiss, Jena, Germany) equipped with a Chameleon Ultra II tunable pulsed laser (690–1080 nm range excitation, Coherent, Santa Clara, CA, USA) and a HM 650V vibratome (Microm Microtech, France) allowing automation of sample cutting and imaging. The images were obtained with a 20x/1.0NA (2.4 mm WD) objective. The instrument was controlled by the ZEN software and a custom ZEN NeCe software extension (Zeiss). Images were acquired through the whole stem base. The imaged sections were aligned under Fiji v2.0.0-rc-53 [17]. For 3D reconstruction, aligned sections were used as an input in Imaris v9.1 (Bitplane AG, Germany).

For LCM, stem bases were harvested in 2 mL ice-cold ethanol:acetic acid 3:1 (v:v) fixative solution, infiltrated thrice under a mild vacuum for 15 min and stored for 20 h in fresh fixative solution at 4˚C. Embedding for dissection was performed as described [18]. For post-embedding RNA quality assessment, stem bases were deparaffinized with three 10 min xylene baths at 4˚C, tissue disruption was done in liquid nitrogen using a TissueLyser II system (Qiagen, The Netherlands) with 3 mm steel beads for 15 sec at 30 Hz, and RNA isolation performed using the Plant RNEasy Kit (Qiagen, The Netherlands). RNA quality was assessed on a 2100 Bioanalyzer with the RNA 6000 Pico Kit (Agilent, Santa Clara, CA, USA), and RINs exceeded 9.3. Six embedded stem bases were cut into 15 µm sections on a Jung RM2055 (Leica Biosystems, Germany) or a Microm HM355S (Thermo Fisher Scientific, Waltham, MA, USA) microtome, and microdissected on a LMD 7000 laser dissector (Leica Biosystems). CR primordia belonging to the three successive rings present in stem bases at this stage were separately collected. Stem cortex tissues were sampled from the same sections where the CR primordia were collected from. Three pairs of two stem base were used as independant biological replicate. For each of the three rings observed in each pair of stem base, all crown root primordia (2~4 per ring) were collected, pooled, and used for RNA extraction in order to gather enough biological material. RNA extractions were performed using the Arcturus PicoPure RNA Isolation Kit (Thermo Fisher Scientific). One-round amplification and cDNA synthesis was done using the GeneChip WT Pico Kit (Applied Biosystems, Thermo Fisher Scientific) at the Transcriptomics platform of Montpellier University Hospital (CHRU Montpellier, France).

## Array hybridization and analysis

All array hybridization steps were performed at the Transcriptomics platform of Montpellier University Hospital (CHRU Montpellier). Array hybridization on Gene Rice (Jp) 1.1 ST chips was done on the GeneAtlas system according to the manufacturer's instructions (Affymetrix, CA, USA). Probe intensities were normalized using Robust Multi-array Average [19] as implemented in Transcriptome Analysis Center software version 4.0.0.25 (Applied Biosystems, Thermo Fisher Scientific). Our dataset was deposited in NCBI's Gene Expression Omnibus [20] and is accessible through GEO Series accession number GSE133593.

## Determination of differentially expressed genes, identification of co-expression clusters, and exploration of ontologies

Differentially expressed genes (DEG) were detected among the 29,664 genes present on the chip using the built-in Transcriptome Analysis Console (TAC) version 4.0.0.25 (Applied Biosystems, Thermo Fisher Scientific) function, using a fold-change cutoff of 1.5 and a $p$-value cutoff of 0.01. Co-expression clusters were identified by performing weighted gene coexpression network analysis (WGCNA) using the consensus method described by [21] on the normalized expression levels of the 3975 differentially expressed genes provided by TAC and for the four tissue types (cortex plus three stages for CR primordia, three replicates) considered as

treatment factors. For our list of DEGs, we used the antilog normalized expression values, and chose a soft-thresholding power of P = 6. Coexpression cluster naming used the WGCNA convention of using color names. MapMan ontologies were assigned to detected DEG according to the ontology mapping BinTree RAPDB-IRGSP1.0 version 1.0 [22]. Granularity of the MapMan bins, that are nested ontologies, was reduced to the second level (e.g. 27.2: RNA.transcription) for readability. The most represented ontologies were identified by considering the ontologies assigned to at least 2% of the total number of DEG, excluding the ontology 35.2 corresponding to "not assigned.unknown and unmapped genes". From the standard, most granular assignment of ontologies to DEGs, a parametric statistical enrichment in ontologies was computed using PageMan as implemented in MapMan v3.6.0-RC1 by performing a bin-wise Wilcoxon test, with Benjamini and Hochberg adjusted $p$-values [23]. Among the DEGs, the genes encoding transcription factors were identified using annotations from PlantTFDB v4.0 [24].

## Results and discussion

### Three rings of CR primordia corresponding to 3 different developmental stages are present in 8 days old rice seedlings

The stem base of rice plants eight days after sowing was imaged to characterize the CR primordia developmental stages that were present. We observed three rings of CR primordia in the stem base (**Fig 1A**). Each ring corresponds to sequential events of CR initiation and therefore represents a different developmental stage [1]. CR primordia belonging to each ring are

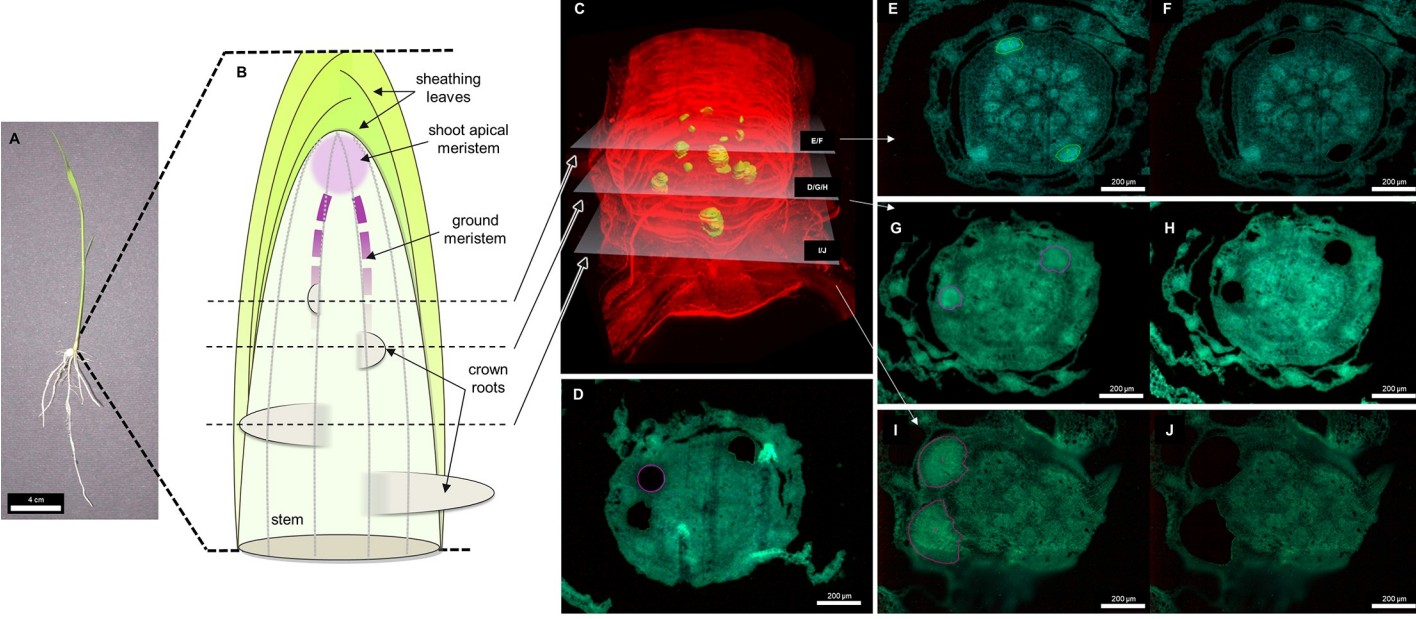

**Fig 1. Visualization of the different developing crown root rings, and the delimitations of sampled tissues.** A: Picture of a rice plantlet used for this experiment and localization of the sampled area; B: Schematic representation of the stem base, crown root initiation area, and positioning of crown root rings; C: Result of 3D block reconstruction from biphoton microscopy of a whole stem base. Root primordia are identified in green, other tissues are in 3D transparency mode. White planes represent the relative positioning of the microtome sections retained for sampling of the early (E, F), intermediate (G, H) and late (I, J) crown root primordia tissues. D: autofluorescence image of a stem base section where cortex tissue was sampled (purple contours). E, F: autofluorescence image of a stem base section before (E) and after (F) laser capture microdissection of early stage developing crown root tissue. G, H: autofluorescence image of a stem base section before (G) and after (H) laser capture microdissection of intermediate stage developing crown root tissue. I, J: autofluorescence image of a stem base section before (I) and after (J) laser capture microdissection of late stage developing crown root tissue.

representative of a given CR developmental stage and were named as previously defined [13], from the top to the base of the stem, early, intermediate and late.

## A progressive shift in transcriptional regulation occurring during crown root formation activates peptide mediated signaling cascades elements and repress specific cell wall metabolism genes

Early, intermediate and late CR primordia, and cortex tissue neighboring the three stages of CR primordia were sampled using LCM from a total of six stem base (**Fig 1B–1H**). Transcriptome analyses were conducted on these different samples. A total of 3975 genes were detected as differentially expressed between cortex and CR primordia of all stages (**S1 Table**, **Fig 2A**).

2208 and 1767 genes were up- or downregulated, respectively. Nearly 30% of these differentially expressed genes (DEGs) were shared between the three developmental stages, and 10.5%, 19.5% and 12.8% of the DEGs were specific of the early, intermediate and late stages, respectively. 10% were shared between early and intermediate, and 15.3% between intermediate and late stages. Only 2% were shared between late and early stages, suggesting a progressive shift in genome expression profile during CR primordia formation. Furthermore, 10 co-expression

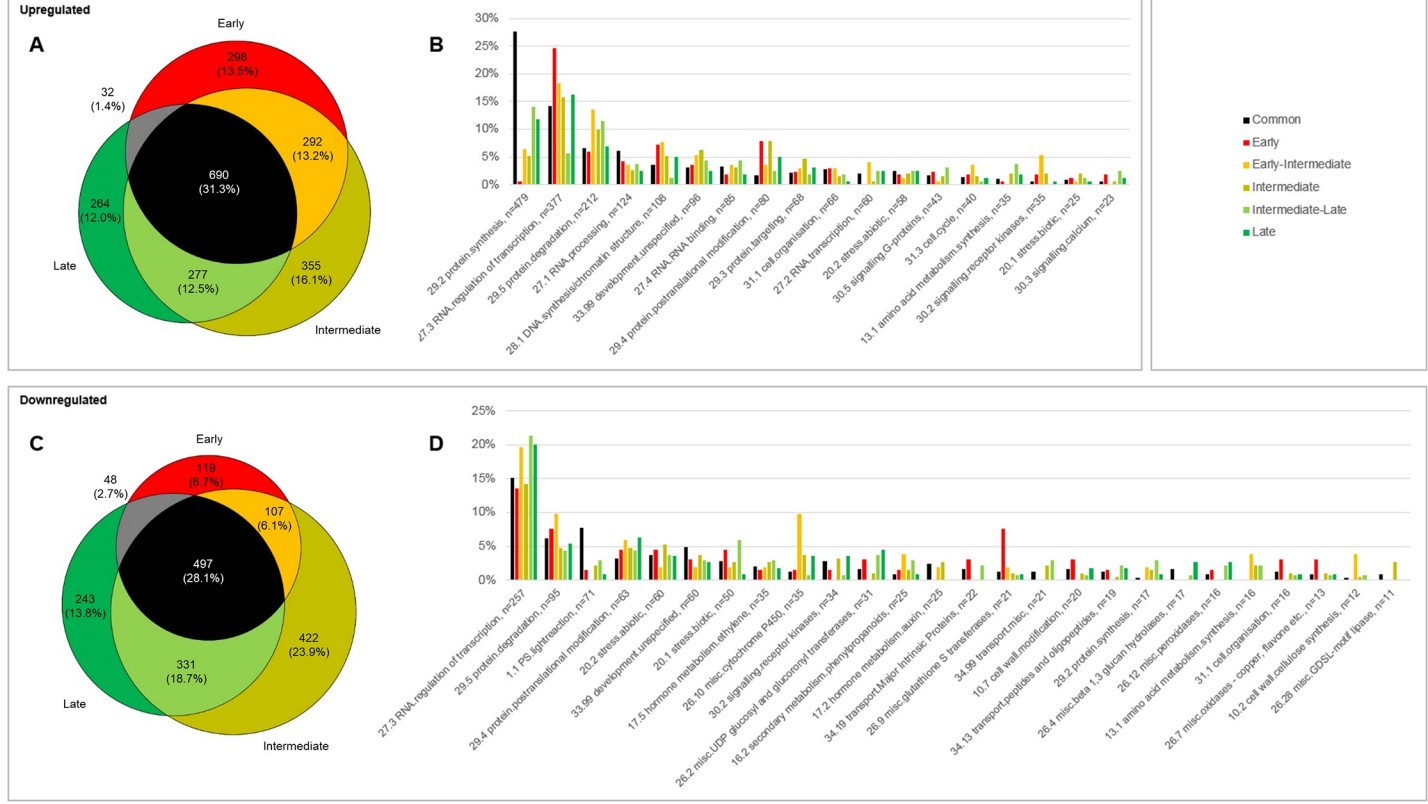

**Fig 2. Repartition of detected DEG in the three developmental stages between the most represented second-level MapMan ontologies.** A: Venn diagram of the number and percentage relative to the total up- or down-regulated DEG of upregulated DEG between the cortex and the early, intermediate and late crown root primordia developmental stages. B: Frequency of mapping of the Venn diagram sections to the most represented MapMan ontologies for upregulated genes. Only the ontologies grouping at least 2% of upregulated or downregulated DEG for one or more Venn segment were plotted. The Late-Early Venn section was not represented. C: Venn diagram of the number and percentage relative to the total up- or downregulated DEG of downregulated DEG between the cortex and the early, intermediate and late crown root primordia developmental stages. D: Frequency of mapping of the Venn diagram sections to the most represented MapMan ontologies for downregulated genes. Only the ontologies grouping at least 2% of upregulated or downregulated DEG for one or more Venn segment were plotted. The Late-Early Venn section was not represented. Ontologies were sorted by decreasing number of DEG (n value).

| UP | | Number of DEG | | | Average FC | | |
|---|---|---|---|---|---|---|---|
| | | Early | Intermediate | Late | Early | Intermediate | Late |
| 29.2 | protein.synthesis | 141 | 168 | 170 | 3.29 | 3.47 | 3.50 |
| 27.3 | RNA.regulation of transcription | 140 | 134 | 103 | 2.80 | 2.78 | 2.61 |
| 29.5 | protein.degradation | 63 | 90 | 59 | 2.06 | 2.15 | 2.16 |
| 27.1 | RNA.processing | 41 | 45 | 38 | 2.75 | 2.88 | 2.71 |
| 28.1 | DNA.synthesis/chromatin structure | 41 | 41 | 26 | 3.35 | 2.70 | 2.72 |
| 33.99 | development.unspecified | 29 | 42 | 25 | 2.88 | 2.36 | 2.56 |
| 27.4 | RNA.RNA binding | 25 | 34 | 26 | 2.33 | 2.40 | 2.64 |
| 29.4 | protein.postranslational modification | 27 | 33 | 20 | 1.94 | 2.00 | 2.08 |
| 29.3 | protein.targeting | 21 | 27 | 20 | 2.66 | 2.56 | 2.58 |
| 31.1 | cell.organisation | 24 | 24 | 18 | 2.27 | 2.38 | 3.97 |
| 27.2 | RNA.transcription | 19 | 21 | 20 | 2.63 | 2.60 | 2.36 |
| 20.2 | stress.abiotic | 17 | 21 | 20 | 2.26 | 2.28 | 2.20 |

| DOWN | | Number of DEG | | | Average FC | | |
|---|---|---|---|---|---|---|---|
| | | Early | Intermediate | Late | Early | Intermediate | Late |
| 27.3 | RNA.regulation of transcription | 61 | 103 | 93 | -2.05 | -2.15 | -2.23 |
| 29.5 | protein.degradation | 29 | 35 | 31 | -1.83 | -1.97 | -2.07 |
| 1.1 | PS.lightreaction | 20 | 27 | 24 | -3.07 | -3.08 | -3.19 |
| 29.4 | protein.postranslational modification | 15 | 26 | 22 | -1.85 | -1.92 | -1.83 |
| 20.2 | stress.abiotic | 15 | 25 | 20 | -3.36 | -2.56 | -3.07 |
| 33.99 | development.unspecified | 16 | 24 | 20 | -1.98 | -1.95 | -1.96 |
| 20.1 | stress.biotic | 12 | 21 | 17 | -4.65 | -2.86 | -2.55 |
| 26.10 | misc.cytochrome P450 | 10 | 16 | 9 | -2.58 | -2.21 | -1.82 |
| 17.5 | hormone metabolism.ethylene | 8 | 15 | 12 | -2.71 | -2.40 | -2.39 |
| 30.2 | signalling.receptor kinases | 8 | 14 | 12 | -1.97 | -1.87 | -1.94 |
| 26.2 | misc.UDP glucosyl and glucoronyl transferases | 6 | 11 | 14 | -2.43 | -1.89 | -1.80 |
| 26.9 | misc.glutathione S transferases | 9 | 7 | 5 | -4.50 | -2.00 | -2.10 |

**Fig 3. Most represented MapMan ontologies for upregulated and downregulated DEG of each stage.** Mapman ontologies sorted by the number of total DEG between the three developmental stages. Only the ontologies representing more than 2% of the total number of DEG per condition were retained.

clusters were identified, characterized by homogeneous changes in expression profiles across samples, and which can be used to explore genes implied in tightly regulated biological processes (S1 Fig).

The most strongly DEGs (detected in 1st and 99th centiles of the observed FC for at least one of the CR primordia developmental stage) are listed in S2 Table. Most of the genes are assigned to transcription or protein synthesis or of unknown function and their role in crown root formation should be further studied. It is interesting to note that *XYLOGLUCAN ENDO-TRANSGLUCOSYLASE/HYDROLASES* (*XTH*) *OsXTH17* gene was among the most downregulated genes. *OsXTH1*, *OsXTH6* and *OsXTH10* were also significantly downregulated in CR primordia compared to cortex (S1 Table).

We next analyzed the ontologies associated with the different stages of development by assigning MapMan ontologies to the DEGs (**Figs 2B and 3**) and statistically tested their enrichment (S2 Fig). Altogether, this showed that the upregulated DEG were more frequently distributed in the protein synthesis, RNA/regulation of transcription, protein degradation, RNA processing, and DNA synthesis/chromatin structure categories, which can be expected from meristematic tissues. Notably, the protein synthesis category (with 89% of mapped genes corresponding to ribosomal proteins) was strongly represented and significantly enriched, indicating a dynamic protein synthesis and regulation in all three developmental stages. Also, we observed a high number of DEGs in the regulation of transcription category (377 up- and 257 downregulated).

Consistently, DEGs assigned to the "33.99 development.unspecified" MapMan ontology are also enriched in transcription factors (TF; S3 Table). Interestingly, the two genes that are the most upregulated at the early stage and that decrease in subsequent stages are *PLETHORA 3/OsPLT3* and *PHYTOSULFOKINE 1 PRECURSOR/OsPSK1*. *OsPSK1* encodes for a sulfated peptide growth factor [25]. Its ortholog in *Arabidopsis* is involved in the regulation of the quiescent center cell number and of the differentiation and cell size of root cap and root meristem cells [25,26]. In *Arabidopsis* roots, maintenance of the quiescent center, meristem, and columella stem cell identity and cell division, is also dependent on TYROSYLPROTEIN

SULFOTRANSFERASE (TPST)-mediated activation of ROOT GROWTH FACTORS (RGF). The transcription factor genes *AtPLT1* and *AtPLT2* mediate these TPST/RGF responses [27,28]. TPST-dependent activation of RGFs was proposed to define the patterns and expression levels of PLTs that, in turn, regulate stem cell niche maintenance [27]. Because *OsPLT1* and *OsPLT3~6* were upregulated in CR primordia (S3 Table) and are in the same phylogenetic group as *AtPLT1* and *AtPLT2* [29], it is tempting to speculate that the *OsPLTs* DEG, and in particular *OsPLT3* that has a similar expression profile with *OsPSK1*, could be involved downstream of a *OsPSK1*-mediated cascade that could contribute to CR primordia formation in rice.

## Several transcription factors involved in root development are differentially expressed during crown root formation

Because of the large proportion of DEG encoding transcription factors in our dataset (S3 Fig), we investigated the TF differentially expressed during CR development associated to root development (Table 1).

Interestingly and adding to the putative involvement of *OsPLT3* in a *OsPSK1*-mediated cascade discussed in the previous paragraph, most of the genes of the PLT family (*OsPLT1*, *OsPLT3~6*) were upregulated in developing CR (Table 1). They are all members of the clade A of the PLT family, and known for their specific expression in both primary root and CR meristem initials [29]. Expression pattern analysis of these genes by *in situ* hybridization showed that they were specifically expressed in different sectors of the CR meristem [29]. This suggests that these 5 TF might be important regulators in different phases of CR primordium development. PLT genes have been demonstrated to play important roles in root branching in Arabidopsis [48]. *AtPLT5*, along with *AtPLT3* and *AtPLT7*, redundantly regulate the formative cell divisions in lateral root primordia and the proper establishment of gene expression programs leading to the establishment of a new growth axis [48]. *AtPLT5* is the closest Arabidopsis phylogenetic relative of *OsPLT3*, *OsPLT4* and *OsPLT5*; as is *AtPLT4* for *OsPLT1* and *OsPLT6* [29]. In lateral root primordia, *AtPLT3*, *AtPLT5* and *AtPLT7* are required for the early and patterned expression of *SHORTROOT* (*AtSHR*) and *WUSCHEL-RELATED HOMEOBOX 5* (*AtWOX5*) and the activation of *SCARECROW* (*AtSCR*), that specify the identity of the ground tissue and quiescent center cells [48]. By contrast, *OsCRL5/OsPLT8*, a member of the clade B of the PLT gene family which encodes an APETALA/ETHYLENE RESPONSE FACTOR (AP2/ERF) and possesses a mutant exhibiting a reduced number of CR due to impaired CR initiation [41], was detected in our study as downregulated during CR primordia development (Table 1). However, this downregulation is coherent with previous RT-qPCR expression profiling experiments where it appeared to be more expressed in stem tissues than in roots [29]. Furthermore, p*OsCRL5*::GFP reporter expression pattern showed expression not only in CR primordia, but also in the stem cortex at the level where CR primordia are initiated [41]. This lack of specificity in expression may explain its detection as downregulated in the three analyzed considered stages of CR development versus cortex tissues (Table 1). This may also indicate that this gene plays a key role during CR initiation but is not involved in later steps of CR primordia differentiation. In any case, further functional characterization of *OsPLTs* genes is needed to clarify their role during CR development.

In Arabidopsis, *AtSHR* and *AtSCR* are activated by *AtPLTs* [48]. Interestingly *OsSCR1* and *OsSHR1* were upregulated in early CR primordia (Table 1). *OsSCR1*, coding for a GRAS-domain TF involved in the definition and stabilization of the quiescent center in primary roots [49] and in the establishment of cortex and endodermal cell lineages [50] is specifically expressed in the endodermis, whereas *OsSHR1* is expressed in the stele. They are involved in

**Table 1. List of differentially expressed transcription factors with a putative function in the development of CR primordia.** CR: Crown root, CK: Cytokinin, GA: Gibberellin, IAA: Auxin, QC: Quiescent center.

| Regulation | TF family | Gene | RAP-DB id | MSU id | Early | Intermediate | Late | Coexpression cluster | Description | References |
|---|---|---|---|---|---|---|---|---|---|---|
| Upregulated | AP2 | *OsPLT5* | Os01g0899800 | LOC_Os01g67380 | 2.06 | 2.19 | | midnightblue | CR formation and development, maintenance of stem cell activity around QC / activation by IAA, repression by CK | [29] |
| | | *OsPLT3* | Os02g0614000 | LOC_Os02g37070 | 7.76 | 7.53 | 3.98 | midnightblue | | |
| | | *OsPLT4* | Os04g0474200 | LOC_Os04g39570 | | 2.31 | | skyblue | | |
| | | *OsPLT1* | Os04g0653300 | LOC_Os04g55970 | 2.74 | 2.9 | 3.08 | skyblue | | |
| | | *OsPLT6* | Os11g0295900 | LOC_Os11g19060 | 2.75 | 2.96 | | skyblue | maintainance of stem cell activity around QC / activation by IAA, repression by CK | [29] |
| | C3H | *C3H13* | Os02g0161200 | LOC_Os02g06584 | 2.58 | | 2.01 | skyblue | coding for BRI1-interacting protein; *bri1* mutant has a mild root phenotype | [30] |
| | ERF | *ERF61* | Os05g0331700 | LOC_Os05g29810 | 5.8 | 4.78 | | skyblue | in cucumber, ortholog is strongly expressed in melatonin-induced lateral root formation | [31] |
| | GRAS | *OsSCR1* | Os11g0124000 | LOC_Os11g03110 | 3.32 | | | midnightblue | *OsSCR1* is specifically expressed in the CR endodermis, whereas *OsSHR1* is expressed in the stele. OsSCR1 and OsSHR1 interact with each other when produced in yeast, similar to SCR and SHR in Arabidopsis. Experimental evidence suggest that *OsSCR1* and *OsSHR1* control the division of the epidermis-endodermis initial cells in rice. | [32] |
| | | *OsSHR1* | Os07g0586900 | LOC_Os07g36820 | 2.44 | | | midnightblue | | |
| | LSD | *OsLOL2/OsLSD5* | Os01g0612700 | LOC_Os01g39710 | 4.94 | 5.34 | 5.14 | skyblue | involved in regulation of GA synthesis | [33] |
| Downregulated | ARF | *OsARF5* | Os02g0138100 | LOC_Os02g04510 | -1.86 | -2.52 | -3.18 | yellow | *ARF5* is downregulated in the *crl6* mutant / *ARF5* in Arabidopsis controls meristem function and organogenesis in both the shoot and root through the direct regulation of *PIN* genes | [34, 35] |
| | | *OsARF24* | Os12g0446370 | LOC_Os12g29520 | | | -1.55 | orange | dimerizes with OsARF23 to promote *RMD* expression, triggering changes in F-actin organization required for auxin-mediated cell growth | [36] |
| | ERF | *ERF99/OsEREBP2* | Os01g0868000 | LOC_Os01g64460 | | -2.2 | -2.14 | yellow | implied in the integration of mechanical signals for root curling through interaction with OsHOS1 | [37] |
| | | *ERF79* | Os02g0594000 | LOC_Os02g34790 | | -2 | -1.74 | yellow | member of the class VIIIb of AP2/ERF family, ortholog of Arabidopsis *DRN*, expressed in center of meristems / *PUCHI*, *DRN*, and *DRNL* interdependently contribute to cellular fate decisions | [38, 39] |
| | | *ERF120* | Os06g0222370 | LOC_Os06g11860 | -1.91 | -2.11 | -2.09 | yellow | direct target of *OsNAC* genes described to alter root architecture | [40] |
| | | *CRL5* | Os07g0124400 | LOC_Os07g02947 | -1.88 | -2.1 | -2.12 | yellow | auxin-induced *CRL5* promotes crown root initiation through repression of cytokinin signaling by positively regulating *OsRR1* | [41] |
| | LBD | *OsLBD3-6* | Os03g0659700 | LOC_Os03g42747 | -1.96 | -2.28 | -3.41 | yellow | downregulated in *osiaa13* mutant / similar to Arabidopsis *ASL18/LBD16*, *ASL16/LBD29*, *ASL20/LBD18* and rice *CRL1/ARL1* | [16, 42] |
| | MYB | *OsMYBS60* | Os12g0124700 | LOC_Os12g03147 | -2.97 | -3.29 | -3.66 | blue | associated with 37–60 cm depth dry weight trait / plays a dual role in abiotic stress responses in Arabidopsis through its involvement in stomatal regulation and root growth | [43, 44] |
| | TCP | *OsTCP7* | Os02g0632800 | LOC_Os02g39347 | -5.9 | -13.86 | -20.34 | yellow | *AtTCP7* in Arabidopsis / in Arabidopsis, overexpression leads to shorter roots; likely to play a role in cell proliferation by restricting G1/S phase transition | [45, 46] |
| | WRKY | *OsWRKY108* | Os01g0821300 | LOC_Os01g60600 | | -2.25 | -1.93 | orange | downregulated in *OsRR6*-ox line, a negative regulator of CK signaling, which showed poorly developed root systems | [47] |

(References cited in table reported here for purpose of reference list generation: Li and Xue 2011 [29], Nakamura 2006 [30], Zhang [31], Kitomi 2018 [32], Xu and He 2007 [33], Xu 2017 [34], Krogan 2016 [35], Li 2014 [36], Lourenço 2015 [37], Chandler 2018 [38], Chandler and Werr [39], Chung et al [40], Kitomi 2011 [41], Kitomi 2012 [42], Takehisa 2012 [16], Phung et al [43], Oh et al [44], Yao 2007 [45], Wang 2014 [46], Hirose 2007 [47]).

quiescent center differentiation and maintenance [49], and they regulate the molecular events establishing the cortex and endodermal cell lineages [50]. Detection of their expression in the early CR developmental stage is coherent with previous observations [51,52] and their role in early stages of lateral root development in rice [16] and Arabidopsis [53]. These data suggest that *OsSCR1* and *OsSHR1* likely play a critical role during establishment of CR primordia in rice.

Other TF involved in hormonal signaling pathways were also differentially expressed (**Table 1**) during CR primordia formation. This is the case of *AUXIN RESPONSE FACTOR* (ARF) *OsARF5* and *OsARF24* and *ETHYLENE RESPONSE FACTOR* (ERF) *ERF99/OsEREBP2* and *OsERF120*. All are known to play a role in general modulation of root architecture and all are downregulated in CR primordia [54], whereas *OsERF61*, which cucumber homolog is involved in lateral root development is upregulated [31]. *LSD1-LIKE ZINC FINGER PROTEIN OsLOL2/OsLSD5*, that is upregulated in CR primordia, is known to be involved in the regulation of gibberellic acid synthesis. Interestingly, a crosstalk between gibberellic acid and auxin is involved in CR emergence and elongation in rice [55,56].

## Genes encoding chromatin remodeling factors are differentially expressed during crown root formation

In addition to genes encoding TF, 14 genes encoding for chromatin remodeling factors were found among the upregulated DEG, suggesting that they also contribute to the progressive shift in transcriptional regulation we observed during CR formation (**S4 Table**). Four of those genes coded for SWItch/Sucrose Non-Fermentable (SWI/SNF) complexes including complex B (SWIB) domain containing proteins, and four other were part of the SBF2 family; both are elements of chromatin structure dynamics contributing to the control of root development [57–60].

*CROWN ROOTLESS 6* (*OsCRL6*) encodes a member of the CHROMODOMAIN HELI-CASE DNA-BINDING (CHD) family protein and is involved in CR initiation during which it contributes to the regulation of meristem cells differentiation [61,62]. It was identified in a co-expression cluster prone to display high expression levels in the three primordia stages, and low in the cortex. Interestingly, most of the *OsIAA* genes are down-regulated in the *crl6* mutant, suggesting that *OsCRL6* regulate auxin signaling pathways [34,61,62]. In our data, a number of *OsIAA* genes were expressed differentially at the 3 different stages, indicating that different auxin signaling elements and pathways are sequentially activated during CR meristem formation and patterning.

## Cross-analysis with other rice root related transcriptome data sets pinpoint key genes involved in *de novo* root formation in rice

In order to identify genes that may have a conserved function between the processes of CR and lateral root formation in rice, we crossed the list of DEG obtained in this study with a list of 204 genes (corresponding to the S5 Table from [16]) associated with lateral root initiation and development, identified from the RiceXpro RXP_4001 root gene expression dataset [16,63,64].

Seventy three DEG of our dataset (43 upregulated, 30 downregulated) crossed with the list of genes suggested to be involved in lateral root initiation and development [16] (**S5 Table**). Among these were genes encoding TF such as PLT family members or cell wall remodeling enzymes such as *OsXTH10* or *EXPANSINS* (*EXP*). *OsXTH10* is involved in the xyloglucan metabolism, that plays a central role in monocotyledon cell wall restructuring [65]. In rice, *XTH* gene expression is tissue-specific and growth stage-dependent [66]. In *Arabidopsis*, auxin

accumulation in the tissues overlying the lateral root primordia induces the expression of a set of cell wall remodeling genes, such as that facilitate the growth of the lateral root through the cortex cells [54]. Similarly, the preferential expression of *OsXTH1*, *OsXTH6*, *OsXTH10* and *OsXTH17* genes in the cortex surrounding developing CR might play a role in the restructuring of cortical cell wall. Regarding the *EXPANSINS*, *OsEXPA2*, *OsEXPA7* and *OsEXPB16* were downregulated in CR primordia in comparison with cortex, and were identified as lateral root initiation- and development-specific [16]. *OsEXPA2* is detected through *in situ* hybridization during the development of the seminal root [67].

Interestingly, most of the shared upregulated genes (33/43) are included in the co-expression cluster III of the RXP_4001 dataset which is suggested to play a role in regulating cell division during lateral root and CR initiation and development, particularly in the apical meristem [16]. These shared upregulated genes included in the co-expression cluster III were mainly assigned to the "midnightblue" (20/33, including *OsSHR1*, previously discussed) and "skyblue" (8/33) coexpression clusters in our study, characterized by a strongest expression level in the early and intermediate primordia stages, respectively. Downregulated genes are included into two other co-expression clusters identified by [16] (16/30 for cluster I including *OsXTH10*, 12/30 for cluster II). In particular, cluster I is proposed to play a specific role in the priming and initiation of the lateral root primordium. Also, 20 out of the 43 upregulated DEG were identified as involved in both lateral root initiation and development, at visible lateral root emergence stage ("overlapping gene" column), such as *OsEXPA5* [16].

This cross-analysis showed that among genes likely involved in lateral root initiation and development, 36% (73/204) were also expressed in developing CR primordia. Over these 73 genes, most of the genes upregulated (33/43) are suggested to play a role in regulation of cell division, and a significant part of the genes downregulated (16/30) could play a role in priming and initiation of lateral roots. This suggests that several genes are involved both in CR and lateral root formation in rice, mostly during the post-initiation and cell proliferation stages.

Also, we compared our dataset with the list of 1753 DEG during the 45 hours following the ectopic expression of *CRL1* in the *crl1* mutant background, which is used to synchronously restore CR formation [14]. Three hundred and twenty-four DEG of our dataset (190 downregulated, 134 upregulated, representing about 8% of the total number of DEG) were induced or repressed for at least one time point during the 45 hours following *CRL1* induction (**S6 Table**). 229/324 of these genes showed expression values that were coherent with the time series, *i.e.* genes upregulated or downregulated in the time series were also upregulated or downregulated in the developing primordia from our experiment, respectively. This suggested that the expression of these genes is likely under the control of CRL1 during CR formation. Interestingly genes encoding TF, chromatin remodeling factors and cell wall remodeling enzymes were retrieved.

## Conclusions

In this study, we used LCM to generate tissue-level transcriptome gene expression atlas at 3 stages of rice CR primordia development between initiation and pre-emergence. Our results reveal a shift in transcriptional and post-transcriptional gene regulation occurring during CR development. Several genes already described for their involvement in CR development were retrieved in our dataset. New genes, such as members of the *PLT* gene family and *OsPSK1*, were also identified suggesting the involvement of peptide signal related pathway during CR formation similar to the pathways already described in Arabidopsis during lateral root formation. This suggest a conservation of some mechanisms during post-embryonic root formation in rice and Arabidopsis. Our analysis also pointed out a differential regulation of several cell

wall remodeling enzyme belonging to *XHT* and *EXP* families during CR formation. Functional analyses will be needed to confirm their implication in this developmental process.

Our dataset constitutes an open valuable resource to identify new genes involved in CR development in rice, in particular via transcriptomic meta-analyses, similarly to what has been done in *Arabidopsis* for lateral root development using the VisuaLRTC tool [7] that revealed new key regulators of lateral root development that escaped mutant screening-based investigations [8]. Similarly to what has been successfully proposed on other species such as cotton for fiber quality [68], this resource will be also useful to help the identification of candidate genes in QTL related to rice root development.

## Supporting information

**S1 Fig. Average non-antilogged expression levels observed in detected coexpression clusters, for each sample type.** Error bars represent the average relative standard deviation calculated from the three biological replicates.
(TIF)

**S2 Fig. Result of statistical enrichment test in MapMan ontologies, performed from the list of DEG.** Color encodes the average FC value for the considered ontology. Bin-wise Wilcoxon test was performed on the DEG between CR primordia and cortex for each developmental stage (FC>1.5, p-value<0.01). Furthermore, Wilcoxon p-values were adjusted according to Benjamini and Hochberg.
(TIF)

**S3 Fig. Number and average fold change of differentially expressed genes belonging to specific transcription factors families detected in the three crown root developmental stages.** Color intensity function of the considered values.
(TIF)

**S1 Table. List of genes detected as differentially expressed between cortex and CR primordia of all stages.**
(XLSX)

**S2 Table. List of genes detected as the most strongly differentially expressed between the crown root (CR) primordia and cortex.** These genes exhibited fold change (FC) values under or above the 1st and 99th centiles, respectively, for at least one developmental stage.
(XLSX)

**S3 Table. List of differentially expressed genes detected between crown root primordia and cortex assigned to the "33.99 development.unspecified" MapMan ontology.**
(XLSX)

**S4 Table. Differentially expressed genes coding for chromatin remodeling factors.**
(XLSX)

**S5 Table. List of differentially expressed genes crossing with a list of genes related to lateral root initiation and development from crown root (S5 Table, Takehisa et al., 2012).**
(XLSX)

**S6 Table. Genes crossing between our LCM dataset and the list of 1753 DEG during the 45 hours following the ectopic expression of CRL1 in the *crl1* mutant background (Lavarenne et al. 2019).**
(XLSX)

## Acknowledgments

We thank the DNA chip facility of the IRB, Montpellier University Hospital, part of the Montpellier GenomiX platform, who processed the Affymetrix chips.

## Author Contributions

**Conceptualization:** Jérémy Lavarenne, Jacques Rouster, Laurent Laplaze, Christophe Sallaud, Mikael Lucas, Pascal Gantet.

**Data curation:** Jérémy Lavarenne.

**Formal analysis:** Jérémy Lavarenne.

**Funding acquisition:** Jacques Rouster, Christophe Sallaud, Pascal Gantet.

**Investigation:** Jérémy Lavarenne, Mathieu Gonin, Pascal Gantet.

**Methodology:** Jérémy Lavarenne, Mathieu Gonin, Antony Champion, Marie Javelle, Hélène Adam, Geneviève Conejéro, Marc Lartaud, Jean-Luc Verdeil.

**Project administration:** Jacques Rouster, Christophe Sallaud, Pascal Gantet.

**Resources:** Jacques Rouster, Christophe Sallaud, Pascal Gantet.

**Supervision:** Antony Champion, Marie Javelle, Hélène Adam, Jacques Rouster, Christophe Sallaud, Mikael Lucas, Pascal Gantet.

**Validation:** Jérémy Lavarenne, Marie Javelle, Jacques Rouster, Christophe Sallaud, Mikael Lucas, Pascal Gantet.

**Writing – original draft:** Jérémy Lavarenne.

**Writing – review & editing:** Jérémy Lavarenne, Antony Champion, Marie Javelle, Jacques Rouster, Laurent Laplaze, Mikael Lucas, Pascal Gantet.

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
