## [Decision Letter · Decision Letter 0]

1 Jul 2020

PONE-D-20-14357

Transcriptome profiling of laser-captured crown root primordia reveals new pathways activated during early stages of crown root formation in rice

PLOS ONE

Dear Dr. Gantet,

Thank you for submitting your manuscript to PLOS ONE. After careful consideration, we feel that it has merit but does not fully meet PLOS ONE’s publication criteria as it currently stands. Therefore, we invite you to submit a revised version of the manuscript that addresses the points raised during the review process.

We look forward to receiving your revised manuscript.

Kind regards,

Kristiina Irma Helena Himanen, PhD

Academic Editor

PLOS ONE

Journal Requirements:

' In addition to institutional funding, this work is supported by the French Agence Nationale de la Recherche (Fondation Agropolis, Program Investissement d’Avenir Labex Agro no. ANR-10-LABX-0001-01, 2016–2018 and Partenariat de Recherche Collaborative Entreprise MASTEROOT no. ANR-17-CE20-0028-01, 2018–2021) and by the CGIAR Research Program (CRP) on rice-agrifood systems (RICE, 2017–2022). J.L. is supported by a CIFRE fellowship (No: 2015/0195) from the Association Nationale de la Recherche Technologique, France, and a financial support from Biogemma, a subsidiary of the seed company Limagrain.'

We note that one or more of the authors have an affiliation to the commercial funders of this research study: Limagrain

Reviewers' comments:

Reviewer's Responses to Questions

**Comments to the Author**

1. Is the manuscript technically sound, and do the data support the conclusions?

Reviewer #1: Yes

Reviewer #2: Yes

2. Has the statistical analysis been performed appropriately and rigorously? 

Reviewer #1: Yes

Reviewer #2: I Don't Know

3. Have the authors made all data underlying the findings in their manuscript fully available?

Reviewer #1: Yes

Reviewer #2: Yes

4. Is the manuscript presented in an intelligible fashion and written in standard English?

Reviewer #1: Yes

Reviewer #2: Yes

5. Review Comments to the Author

Reviewer #1: The main objective appears to be the creation of a resource in form of a gene expression database from 3 time points of crown root development in rice. As such it will be very valuable for others as a reference. Cross-referencing to other possibly related resources has already been made in this manuscript, highlighting commonly identified genes. I suggest the objective of providing a resource rather than providing definite answers should be clearly stated in the introduction. Reading the introduction I was expecting more specific candidate gene experiments.

Figure 1: To me it is not entirely clear what these 3 rings are, possibly because I am not that familiar with this type of micrograph. I would suggest authors provide in addition a schematic drawing of the tissue they sampled to make the design of the sampling clear to all readers.

Fig 2 : I suggest to attempt to give roughly representative sizes to the overlapping and non-overlapping areas. As is now the biggest group (in black) has the smallest area and that is misleading

Tables and Figure axes: the symbol for decimal is a “.” Not a comma.

Reviewer #2: In the ms, the authors have used laser capture microdissection to analyse gene expression during early stages of crown root development in rice. Generally, the ms is fluently written. I have following comments:

Title of the ms: “Transcriptome profiling of laser-captured crown root primordia reveals new pathways activated during early stages of crown root formation in rice” indicates that new pathways are activated. Please, emphasise these more, since many parts of the discussion are about observations known already from other contexts (e.g. lateral root formation).

In the ms, a MapMan tool has been used to analyse differentially expressed genes. This is one tool that can be used, additionally, there are other tools available. It would be good that common terms would be used in the general text instead of the tool-connected terms (e.g. now MapMan bin throughout the text).

Improvement of the Figures 2 and 3 presentation needs to be done to make them clearer.

Apparently, the colours in Fig. S1 sample names mean co-expression clusters. This is not explained at all and the colours are also listed in the tables. Is there any other way to make this more clear for the reader?

Discussion about XTH should be transferred to later text. Now this comes before the gene ontology analysis. General findings need to be described first, then specific discussion about some selected genes.

PLT transcription factor family members are now discussed in two chapters. Please, combine these to the same place.

Discussion about cross-analysis of the DEG data with the already published datasets is difficult to follow since the supplementary tables need to be followed all the time to find out e.g. the clusters’ names and the data of the earlier papers. Please, pick more examples of the genes into the main text to make it easier to follow.

Minor comments:

Line 76: There is an easier way to say the sodium hypochlorite concentration than the mixing ratio with water.

Lines 110-111: Samples from two stem base were pooled and three RNA extractions were performed.

So this means that each sample had three biological replicates? How many primordia were collected from each successive ring in one stem base? Is it two or three like shown in Figure 1? Please, clarify.

Line 303: there are 14 genes in Table S4.

6. PLOS authors have the option to publish the peer review history of their article (what does this mean?). If published, this will include your full peer review and any attached files.

Reviewer #1: No

Reviewer #2: No

---

## [Author Response · Author response to Decision Letter 0]

13 Aug 2020

Responses to reviewers comments

Reviewer #1:

• “The main objective appears to be the creation of a resource in form of a gene expression database from 3 time points of crown root development in rice. As such it will be very valuable for others as a reference. Cross-referencing to other possibly related resources has already been made in this manuscript, highlighting commonly identified genes. I suggest the objective of providing a resource rather than providing definite answers should be clearly stated in the introduction. Reading the introduction I was expecting more specific candidate gene experiments.”

• � Thank you for this remark; we updated the manuscript Abstract and the end of the introduction in order to reflect more the “resource” aspect of this work rather than identification of candidate genes. 

• “Figure 1: To me it is not entirely clear what these 3 rings are, possibly because I am not that familiar with this type of micrograph. I would suggest authors provide in addition a schematic drawing of the tissue they sampled to make the design of the sampling clear to all readers.”

• � Thank you for this suggestion. We reworked the Fig 1 by including a schematic aiming at clarifying the understanding of the sampling localization at the whole plant leavel and inside the stem base.

• “Fig 2 : I suggest to attempt to give roughly representative sizes to the overlapping and non-overlapping areas. As is now the biggest group (in black) has the smallest area and that is misleading”

• � We understand the visualization concern that is raised here; we updated our figure so the respective class sizes in the Venn diagram are now proportional to the number of genes. 

• “Tables and Figure axes: the symbol for decimal is a “.” Not a comma.”

• � We modified the concerned figures.

 

Reviewer #2:

• “Title of the ms: “Transcriptome profiling of laser-captured crown root primordia reveals new pathways activated during early stages of crown root formation in rice” indicates that new pathways are activated. Please, emphasise these more, since many parts of the discussion are about observations known already from other contexts (e.g. lateral root formation).”

• � We understand your concern. However and as you mention it, most observations are already known only in the context of lateral root formation in Arabidopsis; On this base we stress in the discussion the fact that we are observing in the early processes of CR development, rice homolog elements involved in similar pathways already described in LR formation; thus the novelty lies in the fact that the activation of these pathways in the particular context of CR formation was not yet described. In addition our work pinpoint new genes that are difficult to discuss more in absence of further functional information on them and that are likely involved in specific rice CR developmental pathways. We believe that these hypothesis and genes will be further functionally studied by the scientific community and we hope that they will contribute to better understand post-embryonic root formation mechanisms. We did some rearrangement and addition in the text that we hope will clarify this point. 

• “In the ms, a MapMan tool has been used to analyse differentially expressed genes. This is one tool that can be used, additionally, there are other tools available. It would be good that common terms would be used in the general text instead of the tool-connected terms (e.g. now MapMan bin throughout the text).”

• � We thank the reviewer for this comment. Throughout our readings, it appeared that the terms describing the object for different type of ontologies weren’t generic ; furthermore, the generic term “ontology” may be confounded with the Gene Ontology (GO) terms. That is why we at first proposed the use of the MapMan-related term “bin”. However, we understand this issue and modified our manuscript to use the generic term instead. 

• “Improvement of the Figures 2 and 3 presentation needs to be done to make them clearer.”

• � We improved the Fig 2 according to the suggestion reviewer 1 as well as Fig 1. Concerning Fig 3 we have trouble seeing what could improve its readability, please let us know what could be done.

• “Apparently, the colours in Fig. S1 sample names mean co-expression clusters. This is not explained at all and the colours are also listed in the tables. Is there any other way to make this more clear for the reader?”

• � Thank you for this remark. The color name-based nomenclature of coexpression clusters has been proposed as a standard practice for the use of WGCNA-based coexpression clustering (e.g. “MEblue”, “MEgreen” clusters). We chose to keep the proposed names as is so they cannot be confounded with other coexpression cluster names that are used elsewhere in our manuscript, i.e. in the Supp. Table 6 where we refer to the coexpression clusters exposed in Takehisa et al., 2012. Furthermore, the use of such non-standard color names allows to limit the confusion with other colors codes used in the manuscript, such as for early, intermediate and late samplings presented in Figure 2 that use a red-yellow-green colorwheel. However, in order to make this clear, we precised the use of color names as index of coexpression clusters in the Material and methods section of our manuscript. Furthermore, we reorganized the figure so the coexpression clusters appear in decreasing average expression level order. 

• “Discussion about XTH should be transferred to later text. Now this comes before the gene ontology analysis. General findings need to be described first, then specific discussion about some selected genes.”

• � We understand this remark will make the manuscript clearer to read, that is why we moved this part later on the manuscript.

• “PLT transcription factor family members are now discussed in two chapters. Please, combine these to the same place.”

• � Thank you for this remark; this point has unfortunately been raised a couple of times along discussions with co-authors, and we already went back and forth on this point. Here, the first part of the discussion related to peptide signaling revolving around PSK1, for which we observed a similar expression pattern than OsPLT3. Because of its TF nature, we chose to discuss the OsPLT3 matter in the next paragraph in which we review other mapped TFs. As an answer to your suggestion and for more clarity, we reworked the transition between the two contiguous chapters, hoping this will benefit to the understanding of our hypotheses.

• “Discussion about cross-analysis of the DEG data with the already published datasets is difficult to follow since the supplementary tables need to be followed all the time to find out e.g. the clusters’ names and the data of the earlier papers. Please, pick more examples of the genes into the main text to make it easier to follow.”

• � We thank the reviewer for this remark that should help understand the interest of cross-analysis, that is why we added cross-references to genes previously discussed In the manuscript.

• “Minor comments: Line 76: There is an easier way to say the sodium hypochlorite concentration than the mixing ratio with water.”

• � We modified it.

• “Lines 110-111: Samples from two stem base were pooled and three RNA extractions were performed. So this means that each sample had three biological replicates? How many primordia were collected from each successive ring in one stem base? Is it two or three like shown in Figure 1? Please, clarify.”

• � As an answer to this, six stem base were used for microdissection. Three pairs of two stem base were used as inedependant biological replicates. For each of the three rings observed in each pair of stem base, all crown root primordia (2~4 per ring) were collected, pooled, and used for RNA extraction in order to gather enough biological material. These precisions were added in the text. 

• Line 303: there are 14 genes in Table S4.

• � Thank you for noticing this typo that we corrected.

Updated financial disclosure statement

 In addition to institutional funding, this work is supported by the French Agence Nationale de la Recherche (Fondation Agropolis, Program Investissement d’Avenir Labex Agro no. ANR-10-LABX-0001-01, 2016–2018, by the Partenariat de Recherche Collaborative Entreprise MASTEROOT no. ANR-17-CE20-0028-01, 2018–2021) and by the CGIAR Research Program (CRP) on rice-agrifood systems (RICE, 2017–2022). J.L. is supported by a CIFRE fellowship (No: 2015/0195) from the Association Nationale de la Recherche Technologique, France, and a financial support from Biogemma, a subsidiary of the seed company Limagrain. Limagrain provided support in the form of salaries for authors J.L., M.J., J.R. and C.S.. The specific contribution of these authors are mentionned in the ‘author contributions’ section.

Updated competing interests statement

J.L. is supported by a CIFRE fellowship (No: 2015/0195) from the Association Nationale de la Recherche Technologique, France, and a financial support from Biogemma, a subsidiary of the seed company Limagrain. J.L., M.J, J.R. and C.S. are employees of the seed company Limagrain. This does not alter our adherence to PLOS ONE policies on sharing data and materials.

---

## [Editor Report · Decision Letter 1]

24 Aug 2020

Transcriptome profiling of laser-captured crown root primordia reveals new pathways activated during early stages of crown root formation in rice

PONE-D-20-14357R1

Dear Dr. Gantet,

We’re pleased to inform you that your manuscript has been judged scientifically suitable for publication and will be formally accepted for publication once it meets all outstanding technical requirements.

Kind regards,

Kristiina Irma Helena Himanen, PhD

Academic Editor

PLOS ONE

---

## [Editor Report · Acceptance letter]

6 Oct 2020

PONE-D-20-14357R1 

Transcriptome profiling of laser-captured crown root primordia reveals new pathways activated during early stages of crown root formation in rice 

Dear Dr. Gantet:

I'm pleased to inform you that your manuscript has been deemed suitable for publication in PLOS ONE. Congratulations! Your manuscript is now with our production department. 

Kind regards, 

on behalf of

Dr. Kristiina Irma Helena Himanen 

Academic Editor

PLOS ONE